# Ecological and Oceanographic Perspectives in Future Marine Fungal Taxonomy

**DOI:** 10.3390/jof8111141

**Published:** 2022-10-28

**Authors:** Nalin N. Wijayawardene, Don-Qin Dai, Prabath K. Jayasinghe, Sudheera S. Gunasekara, Yuriko Nagano, Saowaluck Tibpromma, Nakarin Suwannarach, Nattawut Boonyuen

**Affiliations:** 1Centre for Yunnan Plateau Biological Resources Protection and Utilization, College of Biological Resource and Food Engineering, Qujing Normal University, Qujing 655011, China; 2Section of Genetics, Institute for Research and Development in Health and Social Care, No: 393/3, Lily Avenue, Off Robert Gunawardane Mawatha, Battaramulla 10120, Sri Lanka; 3National Institute of Fundamental Studies, Hantana Road, Kandy 20000, Sri Lanka; 4National Aquatic Resources Research and Development Agency (NARA), Crow Island, Colombo 00150, Sri Lanka; 5Deep-Sea Biodiversity Research Group, Marine Biodiversity and Environmental Assessment Research Center, Research Institute for Global Change (RIGC), Japan Agency for Marine-Earth Science and Technology (JAMSTEC), 2-15 Natsushima-cho, Yokosuka 237-0061, Japan; 6Research Center of Microbial Diversity and Sustainable Utilization, Faculty of Science, Chiang Mai University, Chiang Mai 50200, Thailand; 7Plant Microbe Interaction Research Team (APMT), National Center for Genetic Engineering and Biotechnology (BIOTEC), National Science and Technology Development Agency (NSTDA), 113 Thailand Science Park, Phahonyothin Road, Khlong Nueng, Khlong Luang, Pathum Thani 12120, Thailand

**Keywords:** culture-independent methods, marine-derived fungi, marine fungal taxonomy, obligate and facultative marine fungi, oceanic currents, upwelling

## Abstract

Marine fungi are an ecological rather than a taxonomic group that has been widely researched. Significant progress has been made in documenting their phylogeny, biodiversity, ultrastructure, ecology, physiology, and capacity for degradation of lignocellulosic compounds. This review (concept paper) summarizes the current knowledge of marine fungal diversity and provides an integrated and comprehensive view of their ecological roles in the world’s oceans. Novel terms for ‘semi marine fungi’ and ‘marine fungi’ are proposed based on the existence of fungi in various oceanic environments. The major maritime currents and upwelling that affect species diversity are discussed. This paper also forecasts under-explored regions with a greater diversity of marine taxa based on oceanic currents. The prospects for marine and semi-marine mycology are highlighted, notably, technological developments in culture-independent sequencing approaches for strengthening our present understanding of marine fungi’s ecological roles.

## 1. Introduction

Fungi are found in a wide range of ecosystems, including virgin forests, freshwater, and saltwater. In these various habitats, fungi exhibit different life modes such as saprobes, endophytes, phytopathogens, parasites, and mycorrhizae. In some adverse environments, fungi have adapted to live in unfavourable conditions, e.g., halophilic taxa and rock-inhabiting species. Fungi are essential for a wide range of habitats and ecosystems, and various ecological and taxonomic studies have been carried out to identify fungal species and their ecological roles [1]. Hawksworth and Lücking [2] listed the species associated with bryophytes, algae, endophytic fungi inside vascular plants, tropical foliicolous and fungicolous fungi, mammalian guts, insect guts, and exoskeletons, on and in rocks, and in deep-sea and ocean sediments to demonstrate the diversity of fungal environments. Nevertheless, little-studied and overlooked life modes have recently been recognized as important groups by ecologists and taxonomists, and these are regarded as crucial groups towards answering the question, ‘where are the missing species?’

The magnitude of fungal species is one of the most debated topics among mycologists. During the last three decades, the total number of global fungal species has been estimated based on different approaches [2,3,4,5]. The ubiquity of fungal taxa, however, is still a question of whether current estimates are precise. As methods to identify taxa improve, estimates of total species will converge as the precision in delimiting species diversity from available data increases. In traditional taxonomy, species identification has been mostly focused on morphology, i.e., the morphological species concept. However, with the advancement of molecular phylogenetic technology, a consolidated species concept is now widely accepted [6].

Marine fungi are a significant fungal group that has been studied for over a century. These fungi dwell on a wide range of substrates, including submerged wood, leaves, sand grains, sediment, seagrasses, sponges, various marine invertebrates, marine vertebrates, plankton, algae/seaweeds, mangrove plants, and possibly artificial materials such as plastics [7,8,9]. The marine environment comprises a broad range of habitats, either in the water column or connected with marine habitats such as deep seawater, oceanic waters, and intertidal habitats, e.g., mangroves, all of which have a wide range of salinity levels [10]. Definitions of marine fungi have been documented, e.g., Johnson and Sparrow [11], Tubaki [12], and Pang et al. [10]. The definition according to Pang et al. [10] is ‘any fungus recovered repeatedly from marine habitats as it is able to grow and/or sporulate (on substrata) in marine environments, it forms symbiotic relationships with other marine organisms; or it is shown to adapt and evolve at the genetic level or be metabolically active in marine environments’. The above definition was accepted in subsequent studies, such as that by Jones et al. [13].

In this paper, we accept the most recent definition of marine fungi since they can exist in different habitats and because some are not host-specific. However, all marine fungi have been reported from different marine ecosystems. Traditionally, only morphological characteristics, e.g., sporulating structures, were used to characterize fungal marine species. Until the 1990s, light and electron microscopy were the major tools used for fungal taxonomy [14,15]. Since 1990, PCR technology has revolutionized fungal taxonomy, and molecular approaches have become the major tool for identifying and classifying marine and other related marine fungi [16,17,18,19]. The study by Spatafora et al. [20] was the first evolutionary study of marine fungi employing Sanger sequencing molecular data, which led the way to further studies employing molecular data for classifying marine fungi. The state-of-the-art molecular approach is multigene phylogenetic analyses, which can be used to identify marine fungi to the species level, as has been accomplished for other groups of fungi. In previous studies, marine fungi have been reported as dominant groups belonging to Dothideomycetes, Eurotiomycetes, Saccharomycetes, and Sordariomycetes [13,21,22,23,24,25,26,27,28,29,30]. Figure 1 and Figure 2 show the distribution of select taxa that have been introduced during 2010–2020 in the above-mentioned classes. The taxa with multigene sequence data have been included in this study to confirm the placement of the marine fungi together with their habitats and geographical regions.

The limitations of culture-based studies have been widely discussed over the last two decades. It has been shown that certain endophytic taxa do not grow in artificial media [31], while some taxa do not produce sporulating structures in nature; hence, it is impossible to determine their phenotypic characteristics with the current culture methods [32]. Furthermore, it is accepted that ‘the majority of the extant fungal diversity produces no distinguishing morphological structures that are visible or describable’; thus, they cannot be connected to any physical specimen, i.e., they have no phenotypes [32]. These species without defining morphological characteristics are referred to as “dark taxa” or ‘‘dark matter” [33,34,35,36,37,38,39]. Currently, mycologists opine that these taxa are the main component of ‘missing taxa’ [40]. Jones [7] stated that studies on marine fungi should be mainly concentrated on ‘unidentified species on a range of substrates, marine-derived fungi isolated from soils, sand, and water, plankton, deep-sea sediments, endo-biota of marine algae, uncultured fungi, and cryptic species’. Over the last two decades, several fungal culture-based studies have been conducted to reveal novel fungal taxa in marine environments and on other organisms, including mangrove substrates isolated from seaweed, algae, sponges, and the decaying wood of maritime salt marsh plants [41,42,43,44,45,46,47,48,49,50]. Culture-independent methodologies (CIM), which are mostly based on genome analysis, have been the mainstay of ecological research since 2005. CIM methodologies such as shotgun and next generation sequencing (NGS), also known as high-throughput sequencing (HTS), have been used to study deep-sea species and fungal taxa in marine sediments [51,52,53,54].

The physical and chemical properties of marine ecosystems are important factors in determining the diversity and distribution of marine biota [55]. Biodiversity occurs at macroscales, namely, major oceanic and pelagic ecosystems [56] which are shaped by macroscale processes of oceanographic conditions such as currents and upwelling [57,58,59], trophodynamics [60,61], coastal physiography [62], biogeography [63,64], and basin topography [64,65]. Since the boundaries of these processes cannot be exactly defined and species-specific tolerance limits exist for these factors, the distribution of marine biodiversity in different marine ecosystems is extremely complex.

In this conceptual paper, we propose the continuation of the term ‘semi marine fungi’ along with the term ‘marine fungi’ based on their occurrence in different oceanic ecosystems. In examining semi-marine ecosystems, we suggest an assessment of the physico- chemical and biological parameters of the environment. Secondly, we consider major oceanic currents and upwelling influences on species diversity on a global scale. We map the type locations of species that have been documented (both culture-based and CIM-based), as well as the oceanic currents and upwelling zones. It is noted that certain areas have a greater number of reported species and that these zones correspond to large marine currents. Relatively few marine species have been reported from regions that, along with several other major oceanic currents, have been extensively studied for fungi in terrestrial environments. We discuss the possible causes of these differences in this part. Thirdly, we predict the regions where we expect a higher diversity of marine taxa based on oceanic currents. Finally, we summarize the locations of extreme marine habitats, along with oceanic currents.

## 2. Coastal, Semi-Marine, and Marine Habitats—How Can They Be Defined?

Mycologists have traditionally studied a wide range of habitats in marine ecosystems, such as mangroves [66], seaweeds [67], seagrasses [68], dead animals [69], drift material washed into the sea and salt marsh plants, open coastal waters, and deep-sea sediments [70,71]. The majority of these ecosystems are substantial sources of organic materials in coastal locations, which are likely to support fungi and have thus been extensively researched [7]. Pang et al. [10] broadly discussed the terms, ‘marine fungi’ or ‘marine-derived fungi’, which was also followed by Jones et al. [13]. These coastal ecosystems, however, are part of different ocean ecosystems that show a wide range of physicochemical and biological properties. Here, we further define marine fungi habitats into ecosystem levels, i.e., pelagic habitats, deep-sea environments, and micro-ecosystems, which are more convenient for marine assessment, management, and conservation [72].

Calabon et al. [73,74] used the term ‘semi-marine environment’ to separate the marine habitats near oceans and that are connected with a terrestrial environment from other marine communities/ habitats. As the biodiversity of large-scale ecosystems appears to be affected by small-scale physical processes such as substratum, storm events, tidal range, oceanic currents, and changes in wave exposure, it appears necessary to differentiate between water-based aquatic habitats and land-based coastal habitats such as mutualism, competition, and predation [75]. Hence, a more precise taxonomy of marine fungal ecosystems is critical for understanding species interactions at various scales, including the habitat and ecosystem levels. The marine environment is divided into two broad categories, i.e., coastal and offshore. A coastal environment includes: (a) some or all of the coastal marine area and the water, plants, animals, and atmosphere above it; (b) all tidal waters and the foreshore, whether above or below mean high water level/line (MHWL), and (c) dunes, beaches, areas of coastal vegetation and associated animals, areas subject to coastal erosion or flooding, salt marshes, sea cliffs, and coastal wetlands, including estuaries [76]. The coastal zone is further divided into at least three distinct but interrelated components, including coastal marine (zone of nearshore currents/littoral zone), semi-coastal marine (foreshore and backshore), and coastal terrestrial. In addition to these three zones, fungal ecosystems can be found in the deep marine zone (offshore zone). Therefore, to be more specific and easier to refer to, we encourage the use of the terms ‘coastal terrestrial ecosystems’, ‘semi coastal marine ecosystems’, ‘coastal marine ecosystems’, and ‘deep marine ecosystems’ to include fungal ecosystems in the supralittoral zone, shore/beach (including intertidal zone), littoral zone, and deep-sea, respectively.

Coastal terrestrial ecosystems located in the supralittoral zone are not directly exposed to water, but they can be influenced by splash and are often exposed to sunlight. However, the permeability of the soil/sand in this zone may be an important element in determining the salinity of the groundwater. Sand dunes are recognized as an example of coastal terrestrial ecosystems. In dunes and beaches, different shrub species such as *Ipomoea pes-caprae* and *Spinifex littoreus* and different *Cactus* species are important hosts for fungi. Moreover, some tree species such as coconut (*Cocos nucifera*) are also affected by splash. Although these trees are associated with mangrove communities, they are not submerged in seawater during high tides. The *Pandanus* species, which are dominant plants in some coastal terrestrial ecosystems, were traditionally not of interest to marine mycologists. However, the prop roots of some *Pandanus* species are submerged in seawater and can thus be regarded as belonging to the littoral zone community (Figure 3 and Figure 4). Therefore, we propose that the *Pandanus* species in coastal areas should be included in the study of marine fungi (Figure 3 and Figure 4).

The term ‘semi-coastal marine ecosystems’ refers to those that exist between the intertidal zone and are influenced directly by tides. These habitats are exposed to the sun during low tides and are inundated during high tides. Semi-coastal marine environments include beach woods, salt marshes, mud flats, estuaries, and lagoons. In traditional marine mycology, mycologists have recognized a wide variety of diverse substrates/hosts associated with fungi in the intertidal zone and the shore/shore zone. Driftwoods, dead sea creatures, limestones, rocks, and bird droppings on beaches have mostly been investigated as potential fungal substrates. In contrast to dunes, i.e., terrestrial ecosystems, semi-coastal ecosystems are often directly influenced by marine processes such as waves and tides, together with water temperature and salinity [77,78,79]. In addition, because of the continuous availability of water, the sediments on the ocean bottom contain a higher amount of moisture than those in terrestrial ecosystems. Marine mycologists have extensively studied mangrove ecosystems in lagoons and estuaries as important habitats. Based on the seawater margin, we suggest a new classification of fungi associated with oceans. We propose to classify mangroves as semi-coastal marine ecosystems. Apart from tides, the salinity of water in estuaries and lagoons has a major impact on the diversity of mangroves and their associated fungi.

Coastal marine ecosystems are herein referred to as exclusively submerged communities in seawater, i.e., in the littoral zone to the deep sea. Coastal marine ecosystems receive sunlight through the water column and, hence, can be considered as productive and diverse areas in the seas. Submerged seagrasses, seaweeds, sponges, live and dead corals, mollusks, and pelagic communities (i.e., nekton) in littoral zones are essential hosts for marine fungi and have been extensively investigated (Figure 3 and Figure 4). Deep marine fungal ecosystems are located beyond the littoral zone; therefore, sunlight is a limiting factor. Sea sediments, whale falls, and hydrothermal vents in the abyssopelagic zone (Figure 3 and Figure 5) are the habitats for fungi in this ecosystem. The aforementioned classification of fungal ecosystems is similar to the previous classifications by Raghukumar [80]. Horizontally, Raghukumar [80] recognized the neritic or coastal habitat and the offshore or oceanic habitat, while vertically, the pelagic and benthic ecosystems were recognized. Though, in nature, ecosystems cannot be separated by clear boundaries; hence, in our proposed classifications, these large-scale ecosystems are categorized as semi-coastal marine ecosystems, coastal marine ecosystems, and deep marine ecosystems. We propose, however, that coastal terrestrial ecosystems be classified as a distinct category that occurs near oceans.

### 2.1. Saline Lakes: Are They Marine Ecosystems?

Jones et al. [13] reported species from saline lakes in their list of marine species. A few species have been described from saline lakes which also have extreme environmental conditions similar to oceanic water, viz, *Aspergillus iranicus*, *A. urmiensis* [81], *Emericellopsis persica* [82], *Neocamarosporium chichastianum* [83], *N. jorjanensis*, *N. persepolisi*, *N. sollicola*, and *Purpureocillium sodanum* [82,84]. Nonetheless, it is questionable whether listing these species as marine species is accurate because some collection sites are surrounded by land. *Neocamarosporium persepolisi* was described by Papizadeh et al. [84]. *Neocamarosporium persepolisi* was reported in the Maharlou saline lake in southern Iran, which is not connected to the ocean. Nonetheless, we can assume that a novel *Neocamarosporium* taxon that morphologically resembles *Camarosporium sensu stricto* is rather likely to be found in salty soil. Saline lakes adjacent to coastal regions, however, can be regarded as another key ecosystem. However, herein, we regard that treating taxa from saline lakes surrounded by land as marine taxa is not appropriate because, from an ecological perspective, they represent a distinct ecosystem.

### 2.2. Coastal Upwelling, Oceanic Currents, and Nutrient-Rich Areas

One of the important scientific issues among biologists of different disciplines is the distribution, dispersal, or migration patterns of different species (that are phylogenetically closely related) or the same species in different geographical locations. Vicariance biogeography is one of these emerging disciplines that has been applied to phylogenetic systematics to explain the distribution of species [85]. Oceanic currents are important in the dispersal of marine environment-dwelling species, as are other abiotic components [57,86]. In addition, Hays [57] described the role of ocean currents in distributing heat and nutrients, which are essential factors for marine biota. Furthermore, it has been reported that the sedimentation of some regions is controlled by current flow [87,88]. The impact of oceanic currents on the distribution patterns of different taxa has been broadly studied [57,89,90,91,92,93]. During the last decade, studies have shown the importance of oceanic cycles on the distribution of marine microorganisms [94,95,96,97,98]. Oceanic currents significantly influence diversity owing to their impacts on nutrient circulation and productivity and microbial dispersal over long distances [59,97].

The geographic locations of marine-derived taxa where they were first recorded are superimposed on the global map with oceanic currents and major upwelling indicated in Figure 6 and Appendix A. Some geographical regions with substantial oceanic cycles and upwelling have been reported to have the greatest fungal diversity, e.g., Southeast Asia. Furthermore, due to the abundance of mangrove vegetation and coral reefs in these regions, the available substrates that can support fungi are increased relative to other regions with lesser oceanic cycling and upwelling.

### 2.3. Relationships with Oceanic Current: What Are the Extensively Studied Fungal Habitats and Existing Habitats?

Fungal occurrence is associated with organic matter [7,99]. Hence, oceanic cycles are relevant to nutrient accumulation in some areas, such as mangroves and reefs. However, how ocean currents impact fungal transport, dispersal, and, more broadly, fungal biogeography remains poorly understood [97]. Ocean surface currents and coastal upwelling are one of the key factors for determining the dispersal patterns of marine organisms [100,101]. As shown by global circulation models, changes in the strength and position of wind patterns affect the paths and intensities of large-scale surface currents and the locations of oceanic fronts [102,103,104,105,106]. Many immotile species and early life stages (such as spores, eggs, and larvae) are transported to their habitats by ocean currents. In addition, currents help many marine organisms reach their feeding, breeding, and nursery grounds. Furthermore, oceanic currents supply optimal environmental conditions (temperature and nutrients) for organisms and their habitats. The distribution of many animal species (especially fish) with the currents has been described, but how currents affect the distribution of marine fungal species is unknown. Since it is possible to transport and distribute marine fungi propagules by ocean currents, the phylogenetic relationships among geographical regions should be explored. Wilson et al. [101] predicted that future changes in the strength of oceanic currents could alter species distributions in the oceans, affecting species compositions in marine fungi and their hosts.

The Gulf Stream (North Atlantic), Kuroshio Current (North Pacific), and Agulhas Current (Indian Ocean) are examples of ocean currents that range from small sub-mesoscale characteristics to massive scale feathers [57]. Under a prevailing wind, the balance of surface wind stress and the Coriolis force due to the spin of the Earth results in near-surface ‘Ekman’ currents, with the net flow in a surface Ekman layer (the upper 20 m) oriented to the right of the wind direction in the Northern Hemisphere and to the left of the wind direction in the Southern Hemisphere [107].

### 2.4. Upwelling

Upwelling, the process in which deep, cold water rises toward the surface, is observed throughout the year in significant coastal upwelling zones or eastern boundary upwelling regions associated with major currents (California Current, Peru-Chile Current, Portugal Current, and Canary and Benguela Currents) [107]. Seasonal upwelling systems occur off the coasts of Somalia, Yemen, Oman, and Sumatra in the South China Sea, the southwest coast of India, and the southern shelves of Australia. Coastal upwelling and oceanic currents, along with coastal regions, are important in maintaining nutrients, e.g., nitrate, phosphate, and silicate, in oceanic waters [108]. Thus, coastal areas (or the continental shelf) are rich in both habitat diversity and overall biodiversity [109]. Mangroves are an important part of many coastal ecosystems, and many studies have been conducted on the fungi found in this environment. Wolanski [110] highlighted the significance of estuaries and their adjacent mangrove forests in controlling tidal hydrodynamics and siltation, both of which contribute to anaerobic conditions in deep sediments.

Upwelling brings nutrient-rich cold waters to the surface layers of the sea, which accelerates planktonic growth [111]. Other ecological components of the marine food web will eventually invade the area as a result of the planktonic acceleration. Marine fungi are also important contributors to the upwelling ecosystem, where they have a range of ecological functions [112,113].

### 2.5. Major Upwelling Systems with Extensive Studies of Marine Fungi

The South China Sea is surrounded by China, Taiwan, the Philippines, Malaysia, Brunei, Indonesia, Singapore, and Vietnam, and it exhibits rich ocean upwelling, mostly in response to the south-westerly summer monsoon [114,115] (Figure 6). The Southern Caribbean coastal upwelling system transports subsurface waters off the coasts of Colombia, Venezuela, and Trinidad and Tobago [116]. In the Southern Caribbean upwelling region, alongshore wind stress occurs throughout the year, reaching a peak during the boreal winter months [117]. Although equatorial upwelling occurs in the Pacific and Atlantic Oceans, it is not as pronounced as in the Indian Ocean. Currents from Portugal induce upwelling around the Iberian coast of Europe [118] and support fungi associations [119]. Based on the findings in [112], the role of fungi in the processing of marine organic matter in the upwelling environment on the coast of Chile shows that the seasonal variation in fungal biomass in the water column reflects their capability for hydrolyzing organic polymers. Therefore, the biomass and activity of fungi in this environment respond to the seasonal cycle of upwelling [112]. In the Benguela upwelling system (Namibia), the results indicate that the phylogenetic diversity of fungi across the redox-stratified marine habitats corresponds to functionally relevant mechanisms that support the structuring of carbon flow from the primary producers in marine microbiomes from the surface ocean to the subseafloor [120].

### 2.6. Major Upwelling Systems Understudied for Marine Fungi

From April through September, the Canary upwelling system is prevalent [117]. The Benguela upwelling system (Nambia) is one of the world’s major upwelling zones. On the west coasts of North and South America, upwelling occurs along the California coast and the coasts of Peru and Chile (Figure 6). There is no consistent upwelling along the West Australian coast, and only sporadic upwelling phenomena have been observed, particularly in the summer [121]. The east coast of Africa has several upwelling regions, namely the Agulhas bank, North Kenya, and Somalia-Oman upwelling systems. The Agulhas upwelling occurs at the continental slope along the entire length of the Agulhas Current [122], the most significant current in the southern hemisphere [123]. In the summer, an alongshore current driven by the southwest monsoon causes upwelling between Somalia and Oman [117]. This upwelling is regarded as the most important western boundary upwelling system worldwide in terms of the volume of wind-induced upwelled water [124]. The western boundary of the East Madagascar Current contributes to upwelling in off-shelf regions of southern Madagascar [125]. The upwelling off the southern tip and the west coast of India [126] and the coastal upwelling off the south coast of Sri Lanka occurs mainly in the summer monsoon [127]. Wind stress causes upwelling on the southern coast of Sri Lanka, which occurs annually as a seasonal phenomenon during the southwest monsoon (June–September) [128]. There have been studies on the upwelling systems associated with marine fungi, such as [112,120,129,130,131,132].

### 2.7. Poorly Studied Regions with Great Potential for Improving Fungal Studies

Biodiversity-rich regions, especially tropical and subtropical regions, harbour an underexplored diversity of fungi [2]. In support of this premise, studies of terrestrial fungi in tropical and subtropical regions have revealed a large number of novel species (Hyde et al. [133], and, regarding the asexually producing species, Wijayawardene et al. [134]). Original descriptions of fungal taxa from marine environments are mainly restricted to Southeast Asia, East Asia (mainly Japan), South Asia (mainly India), Europe, and the USA (Figure 6). However, only countries in Southeast Asia and South Asia and Mexico are categorized as tropical or neotropic, while other widely studied regions are classified as temperate. Furthermore, although a large number of novel taxa in terrestrial habitats in other tropical regions, such as Brazil, have been described annually, this is not reflected by the marine fungi. Facilities, funds, experts/researchers, and interested groups are vital in conducting research studies [135]. Countries in Southeast Asia, East Asia (mainly Japan), South Asia (mainly India), and Europe and the United States have funding and experts in marine fungi, including, notably, Gareth Jones and Kevin Hyde in Southeast Asia. Despite the availability of funding, experts, mycologists, researchers, and other key stakeholders are required to carry out a research project and continue marine fungi-related activities through global collaborations [13]. Consequently, we have noticed that the majority of tropical nations have well-developed research programs to study terrestrial fungi or even freshwater fungi [136,137], but marine fungi are less studied when compared to these ecological fungal groups. For example, there have been published fungal studies performed in Spain, India, and China based on terrestrial and freshwater habitats compared with the marine environment [136,137,138,139].

These facts were emphasized by Jones [7], who mentioned that ‘African mangroves, the coastal areas of Australia and South America’ need to be thoroughly studied. We agree with Jones [7] in that biodiversity hotspots, such as tropical South American countries, e.g., Brazil, Peru, and Argentina, the western coast of Africa, island countries, e.g., Madagascar and Sri Lanka, the Western Ghats of India, and poorly explored coastal regions in Southeast Asia, e.g., the Philippines and Indonesia, are likely to be habitats for a large number of new marine taxa. In addition, most of these regions have been recognized as marine biodiversity hotspots (Figure 6) and are adversely affected by global warming and other anthropogenic activities [140]. It is essential to implement studies to assess the fungal diversity of these regions since most biodiversity studies are only focused on floral and faunal diversity. Consequently, fungal diversity may not be adequately documented before it is irreplaceably lost.

### 2.8. Advancement of Technology to Detect Fungi Inhabiting Extreme Ecosystems

Fungi are ubiquitous and can be found in a wide variety of environments, including extreme environments that are unfavourable for most organisms. Major and extensively studied habitats and the life modes of marine fungi were comprehensively discussed by Jones et al. [13]. The phenotypic characters of fungi are the foundation for the identification of taxa in traditional mycology. However, fungi that do not produce any fruiting structure living in extreme environments could represent a substantial portion of the ‘missing fungi’ that remains an open question for mycologists. Marine fungi have been discovered in extreme environments using both culture-based methodology and CIM; however, CIM may be more fruitful for exploring unknown species, in particular, for identifying taxa that do not produce any phenotypic characters, i.e., fruiting structures.

#### 2.8.1. Ecological Sampling and HTS as a Tool of Integrated Sciences

Traditionally, culture-based approaches have been used to analyze fungi in marine habitats [141,142]. This method, however, could be biased against certain marine fungi that have specific requirements for growth that are not supported by standard nutrient medium. CIM, notably, HTS technology, offers solutions to this fundamental challenge [143,144,145,146,147]. Due to their low cost, large data output, speed, and lack of a requirement for DNA cloning, NGS platforms such as Illumina, Ion Torrent, and Pyrosequencing are important research tools for understanding microbial diversity and community structure in a variety of environments [148,149,150,151,152]. A high resolution of microbial diversity to the species level can be obtained by NGS platforms that target the internal transcribed spacer (ITS) genomic region situated between the small-subunit ribosomal RNA (rRNA) and large-subunit rRNA genes. Hence, ITS sequencing has been recommended as a standard fungal DNA barcode [34,153,154,155]. The analysis and monitoring of fungal communities by ITS sequencing has led to the rapid identification of fungi [156,157]. Although the ITS region has demonstrated strong resolution power, resulting in species discrimination in between most fungal taxa, it does this for a minimal number of genera and rarely for species in the genera most commonly isolated from the marine environment, such as *Aspergillus* and *Penicillium*, and a key disadvantage is its inadequate resolution in closely related species, species complexes involving cryptic species, or metabarcoding studies. As a result, most fungal researchers currently use several DNA markers (the LSU, SSU, COX1, RPB1, RPB2, β-tubulin, MCM7, TEF1-α, γ-actin, ATP6, and CaM genes), and integrative (polyphasic) taxonomy for species delimitation are increasingly being applied for fungal groups where ITS does not provide sufficient precision [4,34,155,158]. Metabarcoding technologies employing HTS techniques have been used to characterize environmental DNA obtained from ecological samples to identify total biological entities, i.e., all the organisms represented [159,160]. Using this approach, the hidden organismic diversity has been elucidated for soil, air, plant shoots, root systems, and deadwood environments [38,50,161,162,163]. HTS has also been used to characterize the diversity of fungi and novel lineages found in aquatic environments (including freshwater and marine water) such as tropical coral, deep-sea compartments, and mangrove sediments [164,165,166,167,168,169]. Furthermore, Vohnk et al. [170] and Zheng et al. [171] used HTS to demonstrate that unculturable fungi are present in seagrasses in the Mediterranean and in aquatic plants in Southwest China, respectively.

#### 2.8.2. HTS Studies of Extreme Marine Habitats

The deep sea (generally referring to oceans greater than 200 m depth) is one of the most mysterious and unexplored extreme marine environments. It covers approximately three-quarters of our planet’s surface area, and its average depth is 3800 m. It is characterized by the absence of sunlight, predominantly low temperatures (<4 °C), and high hydrostatic pressures (up to 110 MPa). Since the first report of fungal isolation from the deep sea [172], many fungi have been isolated and reported from different deep-sea environments, including the deep marine subsurface. There are reports of obligate marine fungi inhabiting the deep sea [173,174,175,176,177], but several facultative fungi have been isolated, the majority of which exhibit similarities to terrestrial species, and these include some new fungal species [178,179,180,181,182,183,184,185,186,187,188,189,190,191,192,193]. Deep-sea research using CIM techniques has evolved rapidly over the last two decades, contributing to our understanding of the diversity and ecological roles of fungi in underexplored extreme marine environments [194]. Furthermore, developments in HTS technologies, in particular, longer sequence reads, have revealed fungal communities in greater detail, including dark taxa lineages known only from sequence data. Xu et al. [52], for instance, used both culture-dependent methodology and HTS approaches to study the fungal communities in sediments obtained from the Southwest Indian Ridge. The culture-dependent method revealed 14 fungal taxa, while HTS revealed 79 fungal taxa. In a similar study, deep-sea sediments from Antarctica were studied using a culture-dependent approach [191] and DNA metabarcoding using HTS [195]. In these studies, only taxa belonging to the genera *Acremonium*, *Penicillium*, and *Pseudogymnoascus* within Ascomycota were obtained using a culture-dependent method. In contrast, metabarcoding sequencing data generated using HTS revealed a diverse and robust fungal community comprising Ascomycota, Mortierellomycota, Basidiomycota, Chytridiomycota, Mucoromycota, and Rozellomycota.

In general, the majority of fungal taxa unearthed by culture-dependent techniques in deep-sea sediments are related to Ascomycota and Basidiomycota, and it is plausible that basal fungal taxa are more difficult to culture in media, especially when employing solid culture media [196]. The advantages of using HTS techniques to evaluate fungal communities in deep-sea habitats include the potential to uncover taxa that are difficult to grow and rare taxa and creating estimates of the abundance of each fungal taxon based on the number of sequences reads.

Recent studies investigating fungal communities in deep-sea environments by HTS techniques are listed in Table 1. These studies have revealed remarkably diverse fungal communities with a high percentage of unassigned operational taxonomic units (OTUs) and amplicon sequence variants (ASVs). In Yap Trench deep sea sediments, 80% of the fungal OTUs observed could not be assigned [169], followed by 64% unassigned OTUs in a study of the deep-sea sediments of the Mediterranean Sea’s canyons [197]. These studies indicate that deep sea habitats (mainly deep-sea sediments that have been examined) harbour mostly unknown fungal communities and a greater diversity of unique fungal lineages (dark taxa) than previously recognized before the application of HTS technologies. In some unique deep-sea habitats, such as deep-sea whale fall ecosystems, the detected unassigned fungal OTUs/ASVs are abundant in the community, indicating a possible essential role of these dark taxa in these ecosystems [198]. Therefore, it is important to reveal the taxonomy of the unassigned fungal groups. However, the ITS region is used as a barcode, which provides insufficient phylogenetic information to fully resolve these fungal communities (Table 1) because of the short length of sequence reads and variability of the ITS region.

Although the ITS region is useful for detecting fungal DNA from environments, including deep-sea sediments [199,200], targeting the more conserved 18S rRNA and 28S rRNA regions may be a more effective method for characterizing dark taxa. Richards et al. [99] used 18S rRNA genes to construct a marine fungal phylogenetic tree, which revealed that previously unknown basal fungal lineages within a novel fungal phylum, including the Cryptomycota [7] and NCLC (novel chytrid-like clade) groups were widespread in deep-sea chemosynthetic environments such as hydrocarbon seep sediments [201], hydrothermal ecosystems [182], and the anoxic sediment surrounding a submarine caldera [202]. Some of the novel unassigned fungal ITS sequences may originate from those basal fungal groups. Hence, it is critical to consider both 18S and ITS sequence data for the study of deep-sea ambient fungi.

PCR-walking from known 18S and 28S rRNA sequences to characterize unknown ITS sequences could be used to obtain a more complete picture of species diversity. Furthermore, culture attempts to recover the dark taxa or single-cell genomics for those basal fungal groups would contribute to our understanding of deep-sea fungi and would expand the publicly available database of taxonomic annotation of unidentified fungi discovered by HTS.

**Table 1 jof-08-01141-t001:** Summary of important HTS studies based on the ITS sequence data of deep-sea habitats since 2015.

Reference	Location/Habitat	HTS Method (NGS Platform)	Total OTUs/ASVs	% of Unassigned Fungi
[195]	Antarctic	Miseq	263 ASVs	27.78%
[203]	South China Sea	llumina	1272 OTUs	8.96%
[198]	whale fall	IonTorrent	107 OTUs	37%
[204]	Magellan seamounts	HiSeq	1662 OTUs	34.70%
[205]	Magellan seamounts	Miseq	1979 OTUs	27.03%
[169]	Yap Trench	llumina	890 OTUs	80%
[206]	Gulf of Mexico	Miseq	4421 OTUs	19.29%
[207]	Mariana Trench	HiSeq	91 OTUs	0.04%
[52]	hydrothermal vent	Hiseq	723 OTUs	37.67%
[197]	canyons of the Mediterranean Sea	Miseq	1742 OTUs	64%
[166]	asphalt seeps	IonTorrent	113 OTUs	14.20%
[32]	the Southwest India Ridge	Miseq	250–300 OTUs	0.02%
[32]	Okinawa trough	Miseq	439 OTUs	2.40%

OTU = operational taxonomic unit; ASV = amplicon sequence variant.

#### 2.8.3. Metabolomics and Its Importance in Chemotaxonomy and Discovering Novel Compounds

Chemosystematics (also known as chemotaxonomy) is a branch of mycology that studies chemical variation in fungal cells and uses chemical characteristics to classify fungal species based on distinct differences and similarities in their biochemical compositions; it is particularly useful in the systematic approach of mycological polyphasic taxonomy [208,209,210].

Metabolomics is defined as the qualitative and quantitative analysis of small molecules (molecular weight of <1500 Da; either primary or secondary metabolites) in a biological sample under specific conditions, and it can provide a powerful approach to metabolic profiling, bridging the gap between genotype and phenotype. Metabolomic technologies have emerged rapidly in fungal research employing analytical methods for chemical separation, such as mass spectrometry (MS), e.g., gas-chromatography mass spectrometry, liquid chromatography-mass spectrometry, tandem mass spectrometry, capillary electrophoresis mass spectrometry, and nuclear magnetic resonance spectroscopy (NMR spectroscopy) or magnetic resonance spectroscopy (MRS). Metabolomics in the fungal kingdom serves as the framework for a new compound discovery pipeline i.e., targeted metabolomics, untargeted metabolomics, and a hybrid technique of targeted–untargeted metabolomics [211,212,213,214]. In addition to classical morphological methods, metabolomics-based chemotaxonomy approaches have been extensively developed and implemented in fungi as a large-scale chemo-taxonomical tool to improve taxonomic classifications and to support the differentiation of fungal species [211,215,216,217]. In recent years, several novel secondary metabolites of potential fungal strains have been isolated, screened for bioactivity, and reported, providing lead diverse natural compounds that can be used in a spectrum of pharmaceutical and medicinal applications [218,219,220].

### 2.9. Overlooked Habitats and Regions of Marine Fungi

The ocean is one of the largest and most diverse biomes on Earth. Recent estimates, all of which are based on indirect methods, reveal that seagrasses harbour fungal communities. Although seagrasses provide critical ecosystem services in coastal environments all over the world, marine fungi associated with seagrasses are poorly understood [221,222]. Furthermore, the polar regions are an understudied location harbouring a great diversity of marine fungal taxa [223]. Several studies have predicted under-explored fungal diversity in biodiversity-rich tropical regions [2,129,135,224,225].

We acknowledge that the majority of biodiversity-rich regions and countries lack scientific funding and expertise for taxonomic studies. Hence, it is difficult to provide accurate and comprehensive identifications in these locations using polyphasic techniques. The National Aquatic Resources Research and Development Agency (NARA) is the major institute for scientific studies of marine biology in Sri Lanka, but it currently acknowledges a limited understanding of the island’s marine fungal diversity [134] (Figure 7; e.g., fungi in seagrasses). To solve the knowledge gap, therefore, it is essential to conduct collaborative projects that could assist with funding, and the collaboration of international specialists is critical.

### 2.10. Concluding Remarks

Jones et al. [13] accepted that there were 1257 obligatory marine species in 539 genera, 74 orders, 168 families, 20 classes, and five phyla. However, Jones [7] stated that a large number of taxa have not been described up to the species level and thus remained as ‘unidentified species’ and Pang et al. [226] stated that 49% of the species from mangrove substrates are unidentified. Jones [7] recognized that a ‘lack of sufficient material for an adequate description’ and a lack of access to the literature to compare with previous studies could be the main reasons for these unidentified species. To establish formal descriptions for these undescribed species, it is crucial to conduct a careful and systematic assessment of existing collections [2,227].

We believe that interdisciplinary research will be important in the future to provide broader and more stable concepts and results from different perspectives. Furthermore, studies that aim to discover, describe, and name novel taxa (i.e., basic sciences) from overlooked geographical regions could be easily integrated with conservation and other industrial applications (where possible).

Molecular techniques will be essential for identifying cryptic species and finding new species in rare genera. The marine genera *Aspergillus*, *Penicillium*, *Arthrobotrys*, *Trichoderma*, *Cladosporium*, *Talaromyces*, *Acremonium*, *Fusarium*, and *Phoma* were identified by Jones et al. [13]. Except for *Talaromyces*, all other taxa are described asexually and have few morphological characteristics that can be used to distinguish species. Consequently, DNA sequences are required for species identification. In addition, species are identified only based on morphology results in polyphyletic genera in traditional taxonomy, e.g., *Phoma*. The first description of the type species ‘might have been too broad and the existence of a continuum in morphological characters ignored’ [7]. Evolutionary research using DNA sequences, on the other hand, confirmed and validated the precise genus boundaries. As a result, taxa with overlapping morphological characteristics need to be re-evaluated by multi-gene phylogenetic analysis. Hence, fungal isolation and DNA extraction from cultures are critical processes in modern mycology, as well as in most research that uses Sanger DNA sequencing to characterize species from a marine environment. However, it should be emphasized that it is still essential to preserve and deposit type materials in fungaria for studying their morphological characters and providing type descriptions (Article 8.4; Shenzhen Code 2018).

Molecular approaches are needed to fill the gap in our understanding of fungal diversity and lineages in the Kingdom of Fungi. Future taxonomic research will need to integrate morphology-based studies with modern molecular methods. Taxonomists currently accept the concept of consolidated species, which is based on polyphasic methods for establishing taxa boundaries [6]. The morphological concept and description of fungi is still strong, and it is essential to recognize voucher specimens (Article 40.1: Publication, on or after 1 January 1958, of the name of a new taxon at the rank of genus or below is valid only when the type of the name is indicated). However, since recent ecological studies employing HTS techniques have highlighted hundreds of dark taxa, future fungal naming without vouchers is a subject of significant debate among taxonomists [38,161,228]. Because naming species is necessary to demonstrate their presence, it is critical to recognize these dark taxa as legitimate species to answer the question, “where are the missing fungi?”

## Figures and Tables

**Figure 1 jof-08-01141-f001:**
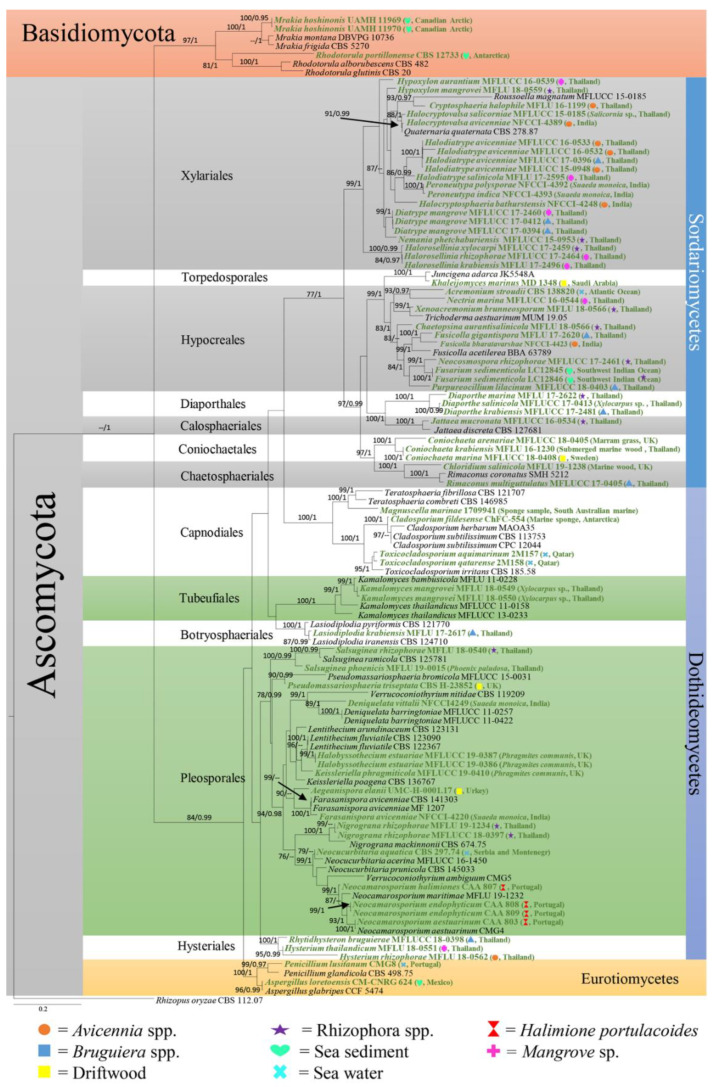
Maximum likelihood analysis with 1000 bootstrap replicates yielded the best tree with the likelihood value of −40790.364548. The combined LSU, *TEF*, and ITS sequence datasets were used for analysis with *Rhizopus oryzae* (CBS 112.07) as the outgroup taxon. The matrix had 1782 distinct alignment patterns, with 76.28% undetermined characters or gaps. The estimated base frequencies were A = 0.242997, C = 0.241567, G = 0.277277, and T = 0.238159; the substitution rates were AC = 1.251989, AG = 2.190388, AT = 1.501627, CG = 1.053941, CT = 4.653615, and GT = 1.000000; and the gamma distribution shape parameter was α = 0.407873. The maximum likelihood bootstrap (ML) values were ≥75% and the Bayesian posterior probabilities (PP) ≥0.95% are given above/below the nodes. The scale bar indicates 0.07 changes. The marine fungi isolates are shown in green and bold.

**Figure 2 jof-08-01141-f002:**
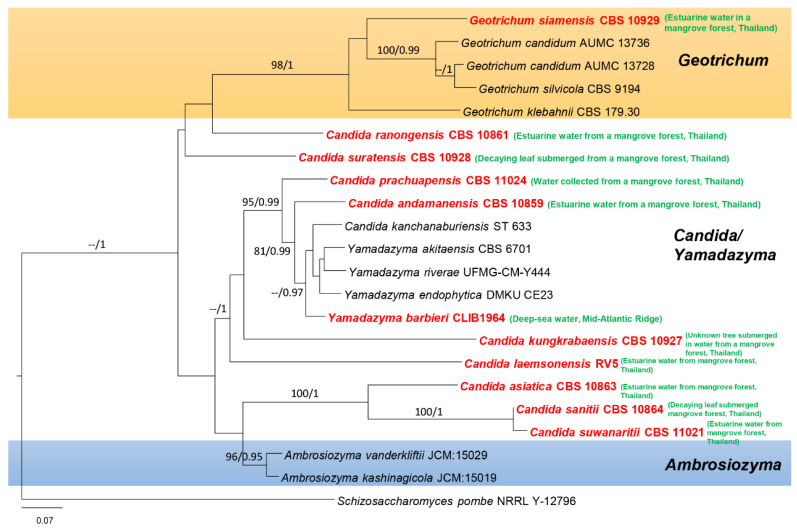
Maximum likelihood analysis with 1000 bootstrap replicates yielded the best tree with the likelihood value of −8256.019037. The combined LSU and ITS sequence datasets were used for analysis with *Schizosaccharomyces pombe* (NRRL Y-12796) as the outgroup taxon. The matrix had 828 distinct alignment patterns, with 50.42% undetermined characters or gaps. The estimated base frequencies were A = 0.277031, C = 0.186671, G = 0.251694, and T = 0.284604; the substitution rates were AC = 0.632253, AG = 1.870647, AT = 1.340467, CG = 0.776808, CT = 4.053598, and GT = 1.000000; and the gamma distribution shape parameter was α = 0.771407. The maximum likelihood bootstrap (ML) values were ≥75% and the Bayesian posterior probabilities (PP) ≥ 0.95% are given above/below the nodes. The scale bar indicates 0.07 changes. The marine fungi isolates are shown in red and bold.

**Figure 3 jof-08-01141-f003:**
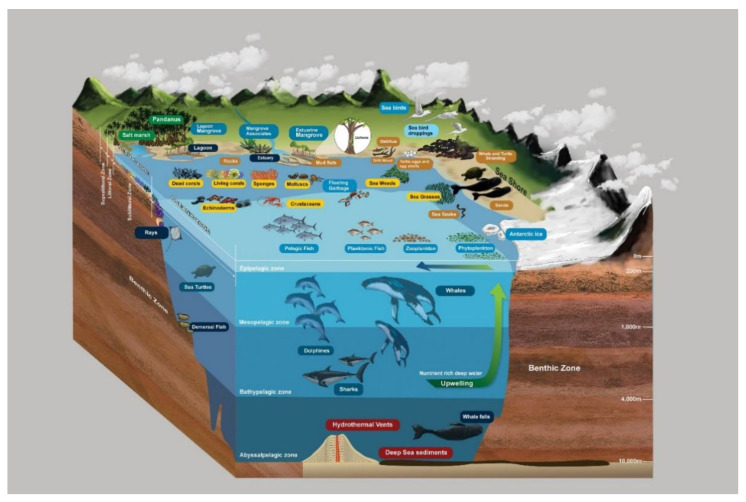
Different habitats and ecosystems for marine fungi.

**Figure 4 jof-08-01141-f004:**
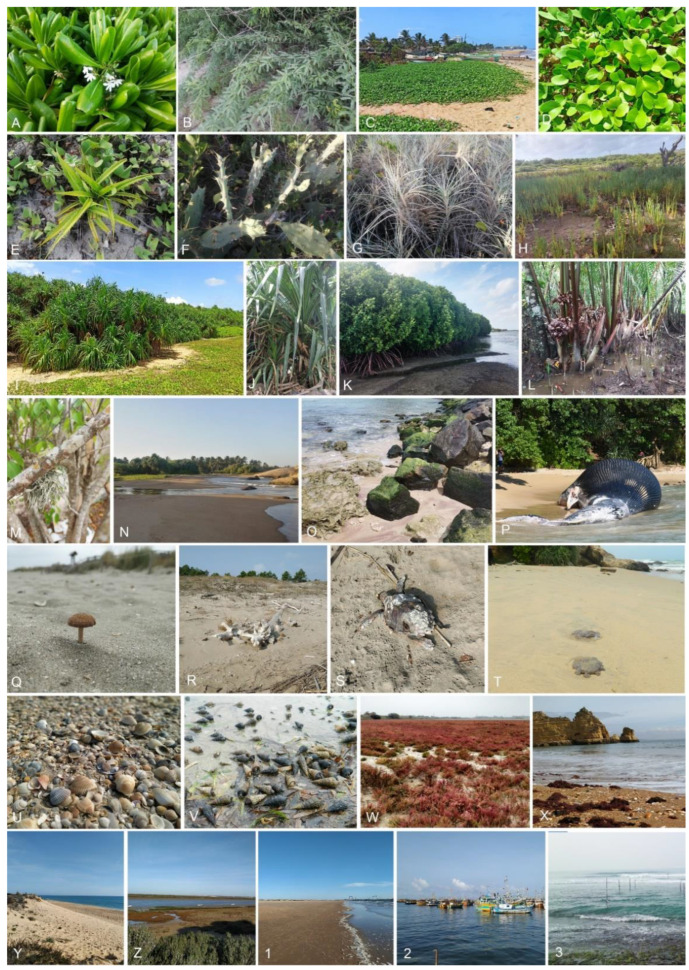
Different habitats hosting fungal marine fungi: (**A**–**J**,**M**), terrestrial-coastal ecosystems. (**K**,**L**,**N**–**1**), semi-coastal marine ecosystems. (**2**,**3**), coastal marine ecosystems. Credit: pictures (**C**,**D**,**I**), M.P. Hendawitharana.

**Figure 5 jof-08-01141-f005:**
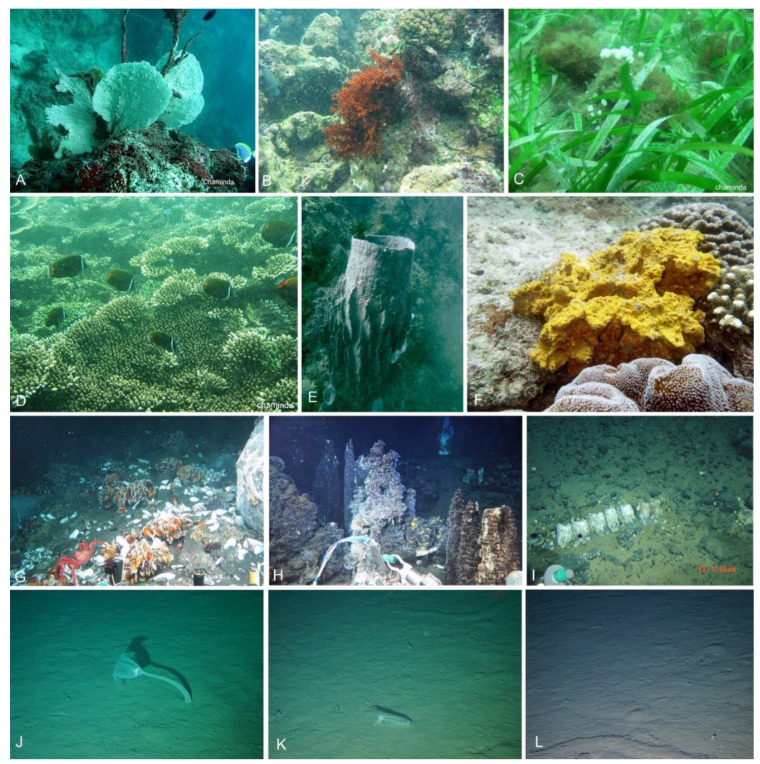
Coastal marine and deep marine habitats. (**A**–**F**), coastal marine ecosystems. (**G**–**L**), deep marine ecosystems. Credit: pictures (**A**–**D**), M.M.C. Karunarathne; pictures (**E**,**F**), P. Cárdenas.

**Figure 6 jof-08-01141-f006:**
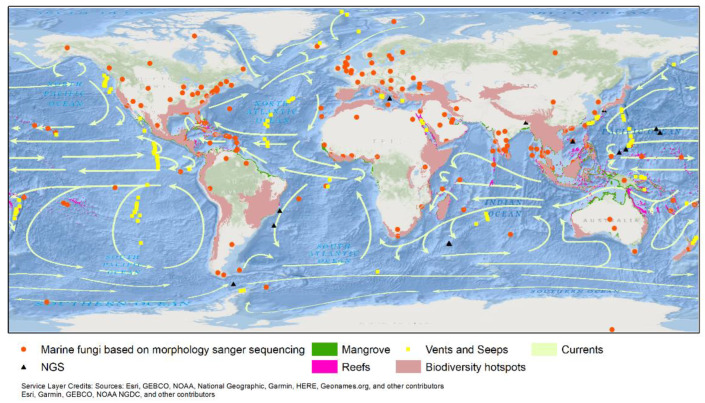
Distribution of marine species based on the Index Fungorum, accessed on 12 September 2022; www.marinefungi.org, accessed on 10 September 2022; and oceanic currents.

**Figure 7 jof-08-01141-f007:**
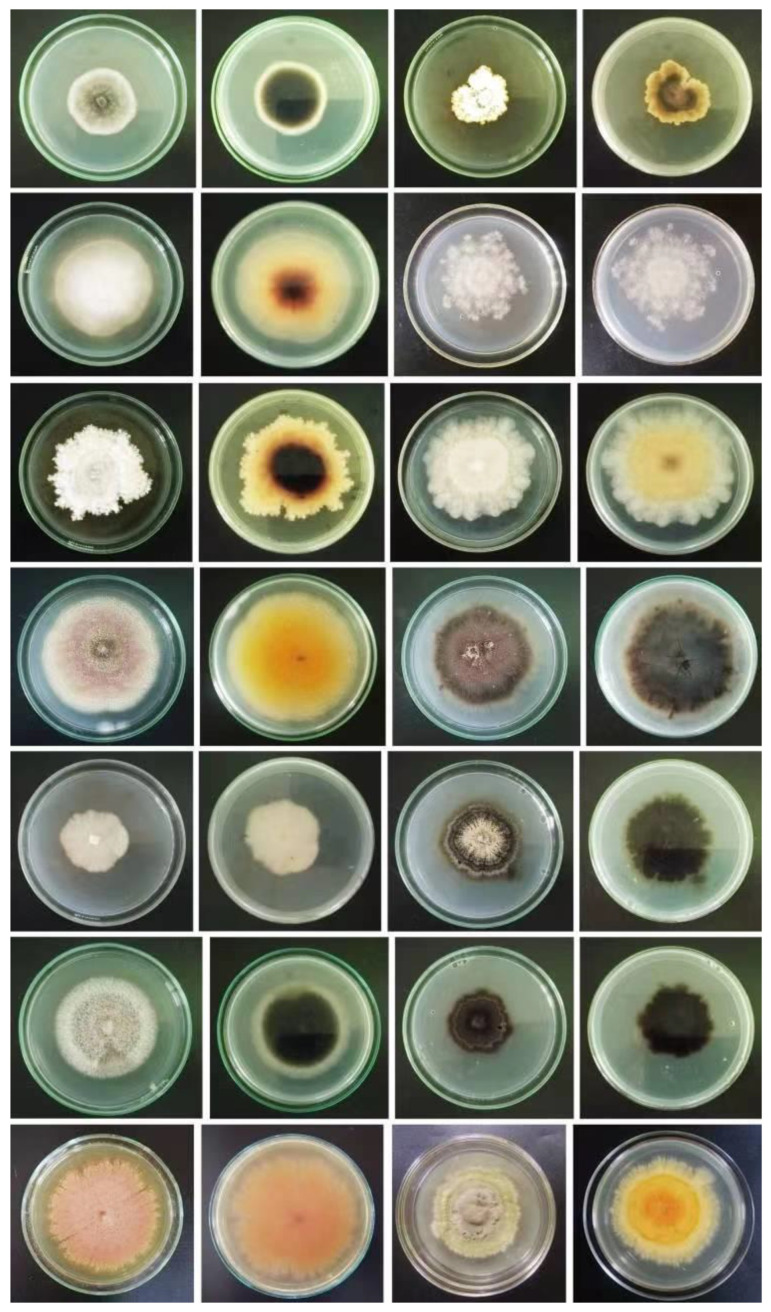
Endophytic and epiphytic fungi isolated from four seagrass species at Kalpitiya lagoon, Sri Lanka (molecular characterization of these taxa is still being carried out faithfully by Rajakaruna et al., though unpublished) (courtesy of Oshadhi Rajakaruna, Faculty of Science, University of Colombo, Sri Lanka).

## Data Availability

All newly generated sequences have been deposited to the Gen-Bank.

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
