# Peer review of "Ecological and Oceanographic Perspectives in Future Marine Fungal Taxonomy"

_jof, 2022, doi:10.3390/jof8111141_

Round 1

Reviewer 1 Report

It is a good manuscript to the aspect of marine fungi.  It would be better if authors could present progresses in marine fungal genomic studies.  Genome-resolved analysis could provide more about fungal taxonomy, such as metabolic potentials which might be more relevant to ecological roles. 

Author Response

Dear Editor and Reviewers,

Thank you for your letter and comments on our manuscript entitled “Ecological and oceanographic perspectives in future marine fungal taxonomy (Manuscript ID: jof-1940225)”.

Your comments and those of two reviewers were highly insightful and enabled us to greatly improve the quality of our manuscript (jof-1940225). We have studied these comments carefully and have made corrections.

All revisions based on all valuable suggestions of the reviewers on the manuscript were revised, recorrected and marked up using “yellow highlighted” in the manuscript file.

Furthermore, all responses to comments from two reviewers are indicated with ">>>>" in red and yellow texts in this file. We hope that the revisions in the manuscript and our accompanying responses will be sufficient to make our manuscript suitable for publication in Journal of Fungi (JOF).

Best regards,

Nattawut Boonyuen, PhD.

=============================================

Reviewer I:

It is a good manuscript to the aspect of marine fungi.  It would be better if authors could present progresses in marine fungal genomic studies.  Genome-resolved analysis could provide more about fungal taxonomy, such as metabolic potentials which might be more relevant to ecological roles.

>>>>Thank you so much for the Reviewer I, this review highlights current knowledge on marine fungal diversity and provides an integrated and comprehensive overview of their ecological significance in the earth's oceans.  We also provide advancement of technology to detect fungi using genomic studies under the HTS technology as shown in pages 13-15.

=============================================

Reviewer II:

In this manuscript, the authors provide a general update on the status of marine mycology without going into details of the currently known marine fungal biodiversity. The article is presented as a ‘concept paper’ highlighting the current technical state of the art of marine mycology in terms of biodiversity and research techniques currently implemented. In this manuscript is presented a new definition for marine areas often subjected to sampling for microbial isolation, in the effort to normalize the location of microbes in future studies and support geographical understanding of marine fungal biodiversity. Authors also analyze oceanic currents worldwide, they highlight areas where high biodiversity was recorded and indicate other areas where few studies were performed, providing suggestions on where additional research should be carried out.

The manuscript is well written and easy to read, the literature is up to date and the information reported on fungal biodiversity in relation to Oceanic currents are interesting. However, I have a series of concerns about several topics reported in the manuscript, and therefore suggest thorough revision before submission.

Abstract

 The abstract is of appropriate length and well written. I made some comments for the sentence ‘diversity of marine fungi in the world’s oceans and provides an integrated and holistic view of their ecological roles’. See PDF file

>>>> We have revised as suggested in Page 1 as follows: “………..the current knowledge of the marine fungal diversity and provides an integrated and comprehensive view of their ecological roles in the world’s oceans”.

  1. Introduction

 In the introduction, authors provide a nice update on the history of marine fungal research, the methods involved in their identification, their definition, biodiversity and sites of isolation. The space dedicated to this paragraph and the figures are appropriate. I only made a couple of comments in the initial part of the text and in the figure one, where typographic errors are present. See PDF file.

>>>> We have recorrected and revised as suggested in Page 2 as follows: “…..Hawksworth and Lücking [2] listed the species associated with bryophytes, algae, endophytic fungi inside vascular plants, tropical foliicolous and fungicolous fungi, mammalian guts, insect guts, and exoskeletons, on and in rocks, and deep-sea and ocean sediments to demonstrate the diversity of fungal environments”.

>>>> We have recorrected and revised as suggested in Page 2 as follows:“………the total number of global fungal species has been estimated based on different approaches.”

>>>> We have revised as suggested in Page 2 as follows:“……… However, all marine fungi are reported from different marine ecosystems”.

  1. Coastal, semi-marine, and marine habitats – how can they be defined?

 In the second paragraph on this paper, the authors propose new definitions for marine habitats where fungi have been documented, in the attempt to better categorize future studies and direct research on areas previously not considered marine (i.e., Pandanus).

The definition and delimitation of specific marine areas for microbial isolation is important to advance understanding on worldwide fungal distribution and avoid misunderstanding in future studies. Therefore, agreement on consensus definitions is essential. However, the proposed definitions suggested by the authors might not fully support their purpose as they are described with terms not specific enough and are characterized by partial overlapping.

The suggested definitions are:

  1. ‘costal terrestrial ecosystems’ covering the supralittoral zone
  2. ‘semi coastal marine ecosystems’ covering shore/beach including intertidal zone (areas “directly influenced by tides”)
  3. ‘coastal marine ecosystems’ covering the littoral zone
  4. ‘deep marine ecosystems’ covering the deep sea

>>>> We have revised as suggested in Page 5 as follows:“………  in the supralittoral zone are not direct exposure to water, but can be influenced by splash and the soil often expose to the sunlight.”

Overlap between the proposed ‘costal terrestrial ecosystems’ and ‘semi coastal marine ecosystems’ areas is associated with the words “shore/beach”. According to CoastalWiki, a widely accessed public webpage curated by the Flanders Marine Institute, the beach or shore comprises the space from the mean low point of water tide to the line of permanent vegetation. This area includes the supralittoral zone fully or in part. In my opinion, the issue could be overcome by eliminating ‘shore/beach’ from the definition and only leaving the intertidal zone, as anything above its higher margin that is influenced by marine environment is considered supralittoral.

>>>> Thank you so much for this issue, the main concern of the reviewer is overlapping of ecosystems. It is true, in nature clear boundaries for ecosystems cannot be found. Since distinct factors to separate the ecosystems, and we have removed "shore" in describing 'semi coastal marine ecosystems' as shown in Page 6 as follows: “…….to those that exist between the intertidal zone and are influenced directly by tides. These habitats expose to the sun during low tides and get inundated during high tides. “…………environments include beach woods, ……”

Also, we added some texts to improve as following here.

>>>>We have revised as suggested and added some key sentence in Page 6 as follows:“………that exist between the intertidal zone and the shore and- are influenced directly by tides. These habitats expose to the sun during low tides and get inundated during high tides”.

>>>> We have revised as suggested and added some key sentence in Page 6 as follows:“……… In addition, because of the continuous availability of water, the sediments on the ocean bottom contain a higher amount of moisture than those in terrestrial ecosystems”.

The proposed ‘coastal marine ecosystems’ zone present an issue in its definition. In this manuscript, it is initially indicated as the area covering the littoral zone, and in a second paragraph indicated as “exclusively submerged communities in seawater”. The encyclopedia Britannica, in the section dedicated to marine ecology, define the littoral zone as the “marine ecological realm that experiences the effects of tidal and longshore currents”, including the intertidal subzone. In this case, presenting overlapping with the ‘semi coastal marine ecosystems’ area. The definition of this area should be more specific in my opinion.

>>>> We have revised as suggested and added some key sentence in Page 6 as follows: “……Coastal marine ecosystems receive sunlight through the water column and, hence, can be considered as productive and diverse areas in the seas. Submerged seagrasses,…….…..”

>>>> We have revised as suggested and added some key sentence in Page 6 as follows: “…….. are located beyond the littoral zone; therefore, sunlight is a limiting factor. Sea sediments, whale falls, and hydrothermal vents in the abyssopelagic zone (Figs 3 and 5) are the habitats for fungi in this ecosystem”.

Ultimately, the difference between ‘coastal marine ecosystem’ and ‘deep marine ecosystem’ is not very clear: the definition of a specific depth or other characteristics to differentiate between the two zones are not indicated. The authors indicate “sea sediments, whale falls, and hydrothermal vents in the abyssopelagic zone” as habitats within the ‘deep marine ecosystem’, however, many other habitats are present in the abyssopelagic zone. Moreover, it is not very clear how should be classified the offshore deep-sea water column.

>>>> We have revised as suggested and added some key sentence in Page 6 as follows: “……..Though, in nature, ecosystems cannot be separated by clear boundaries; hence,…….”

Saline lakes; are they marine ecosystems?

The paragraph reads well and has meaningful content. It is not clear what is the authors’ ultimate opinion on the classification of saline lakes and marine ecosystems.

>>>> We have revised and added additional information as shown in Page 7 (the last paragraph) as follows:“……..Marine fungi and marine-derived fungi exist widely in marine ecosystems, particularly in saline lakes. This habitat is of great importance to studying their species composition, diversity, and relationships with environmental factors to protect and maintain ecosystem balance [13]. Furthermore, salt lakes are essential, biologically and mineral resource-rich lakes that have significant research value and occur on all continents, including Antarctica, the Caspian Sea, the Dead Sea, and many of the highest lakes, such as those in Tibet and on the Altiplano of South America [13].

>>>> We have revised and added additional information as shown in Page 7 as follows: “……..However, herein, we regard that treating taxa from saline lakes surrounded by land as marine taxa is not appropriate because they represent a distinct ecosystem in ecological perspective”.

Coastal upwelling, oceanic currents, and nutrient-rich areas

The paragraph reads well and has meaningful content. Authors indicate some geographical locations with oceanic currents and major upwelling as the greatest for fungal biodiversity (e.g., Southeast Asia). It would be helpful to provide data on the number of samples processed or studies performed in each reported world region to understand if the number of species described is influenced by a larger number of studies performed in a specific area.

>>>> In our dataset, we considered the number of species recorded in a location, rather than the number of publications from the country/region. The summary of number of species recorded per country/region can provided as a supplementary table (see attached file)

Major upwelling systems with extensive studies of marine fungi

Articles reporting fungal biodiversity from these areas are not discussed in the paragraph, in my opinion they should be included and discussed in association with the presented upwelling systems.

>>>> We have revised and added additional information as shown in Page 11 (the last paragraph) as follows: “…….. Based on the findings [219], the role of fungi in the processing of marine organic matter in the upwelling environment in the coast of Chile shows that the seasonal variation in fungal biomass in the water column reflects their capability to hydrolyze organic polymers. Therefore, the biomass and activity of fungi in this environment respond to the seasonal cycle of upwelling [219]. In the Benguela Upwelling System (Namibia), the results indicate that the phylogenetic diversity of fungi across redox-stratified marine habitats corresponds to functionally relevant mechanisms that support in structuring carbon flow from primary producers in marine microbiomes from the surface ocean to the subseafloor [220].

Major upwelling systems understudied for marine fungi

It is not clear if the areas here described were ever studied for fungal biodiversity. If data are present, the studies should be here cited.

>>>> We have revised as suggested and add more citations in Page 11 as follows: “There are few studies on the upwelling systems associated with marine fungi, such as citations here [219–222]”.

Poorly studied regions with great potential to improve fungal studies

Two comments are reported in the pdf file for sentences not easily understandable.

>>>> We have revised as suggested and make more easily to understand in Page 12 as follows:“……...Despite the availability of funding, experts, mycologists, researchers, and other key stakeholders are required to carry out a research project and continue marine fungi-related activities through global collaborations”.

>>>> We have revised as suggested and make more easily to understand in Page 12 as follows: “……...but marine fungi are less studied when compared to these ecological fungal groups, e.g., in Australia.”.

  1. Ecological sampling and HTS as a tool of integrated sciences

In my opinion, this subtitle does not reflect the written content. It is not clear to me what the authors intends as “tool of integrated sciences” for ecological sampling and HTS.

In this paragraph, authors highlight that “High resolution of microbial diversity to the species level can be obtained by NGS targeting the internal transcribed spacer (ITS) genomic region”. Even though the reported literature is relevant, there are numerous studies highlighting the limitation of ITS region to discriminate between different fungal species (see, for example: https://doi.org/10.4137/EBO.S653; or https://doi.org/10.1128/AEM.00626-21; or https://doi.org/10.1038/s41579-018-0116-y). In my opinion, it is important to discuss this limitation in the submitted manuscript.

>>>> We have revised as suggested and added more discussion in Page 13 as follows: “…….. Although the ITS region has demonstrated strong resolution power, resulting in species discrimination in the most of fungal taxa, a key disadvantage is its inadequate resolution in closely related species, species complexes involving cryptic species, or metabarcoding studies. As a result, secondary DNA barcodes are increasingly being applied for groups where ITS does not give sufficient precision [144,146].

  1. Metabolomics and its importance in chemotaxonomy and discovering novel com- pounds

This paragraph is quite confusing and does not read well. Chemotaxonomy and compounds discovery are discussed intermittently and not properly defined. I suggest adding a clear definition for chemotaxonomy and metabolomics, along with their use in fungal taxonomy, biotechnology or biodiscovery. Moreover, the sentence reporting technologies used in metabolomics should be written better (abbreviations in brackets and ‘-’ added where needed).

>>>> In the text file, we already given the definition of chemotaxonomy in Pages 14-15 as follows: “…This is a branch of mycology that studies chemical variation in fungal cells and uses chemical characteristics to classify fungal species based on distinct differences and similarities in their biochemical compositions; it is particularly useful in the systematic approach of mycological polyphasic taxonomy”.

>>>> We have revised as suggested in Page 15 “…….and quantitative analysis of small molecules (molecular weight < 1500 Da;……”

>>>>Also, we have revised the abbreviations in brackets and changed them as the full name in Page 15 as follows: “………. such as Mass spectrometry (MS) e.g., gas-chromatography mass spectrometry, liquid chromatography-mass spectrometry, Tandem mass spectrometry, Capillary electrophoresis mass spectrometry…”

>>>> We have revised in Pages 15-16. Additional discussion was added, and a section (concluding remarks) was switched to the bottom.

>>>> Based on the whole paper (Pages 1-27), a few typo errors have already recorrected and re-double checked, and related citations were included as yellow

Reviewer 2 Report

Review to: Ecological and oceanographic perspectives in future marine fungal taxonomy

In this manuscript, the authors provide a general update on the status of marine mycology without going into details of the currently known marine fungal biodiversity. The article is presented as a ‘concept paper’ highlighting the current technical state of the art of marine mycology in terms of biodiversity and research techniques currently implemented. In this manuscript is presented a new definition for marine areas often subjected to sampling for microbial isolation, in the effort to normalize the location of microbes in future studies and support geographical understanding of marine fungal biodiversity. Authors also analyze oceanic currents worldwide, they highlight areas where high biodiversity was recorded and indicate other areas where few studies were performed, providing suggestions on where additional research should be carried out.

The manuscript is well written and easy to read, the literature is up to date and the information reported on fungal biodiversity in relation to Oceanic currents are interesting. However, I have a series of concerns about several topics reported in the manuscript, and therefore suggest thorough revision before submission.

Abstract

The abstract is of appropriate length and well written. I made some comments for the sentence ‘diversity of marine fungi in the world’s oceans and provides an integrated and holistic view of their ecological roles’. See PDF file

1. Introduction

In the introduction, authors provide a nice update on the history of marine fungal research, the methods involved in their identification, their definition, biodiversity and sites of isolation. The space dedicated to this paragraph and the figures are appropriate. I only made a couple of comments in the initial part of the text and in the figure one, where typographic errors are present. See PDF file.

2. Coastal, semi-marine, and marine habitats – how can they be defined?

In the second paragraph on this paper, the authors propose new definitions for marine habitats where fungi have been documented, in the attempt to better categorize future studies and direct research on areas previously not considered marine (i.e., Pandanus).

The definition and delimitation of specific marine areas for microbial isolation is important to advance understanding on worldwide fungal distribution and avoid misunderstanding in future studies. Therefore, agreement on consensus definitions is essential. However, the proposed definitions suggested by the authors might not fully support their purpose as they are described with terms not specific enough and are characterized by partial overlapping.

The suggested definitions are:

1.     ‘costal terrestrial ecosystems’ covering the supralittoral zone

2.     ‘semi coastal marine ecosystems’ covering shore/beach including intertidal zone (areas “directly influenced by tides”)

3.     ‘coastal marine ecosystems’ covering the littoral zone

4.     ‘deep marine ecosystems’ covering the deep sea

Overlap between the proposed ‘costal terrestrial ecosystems’ and ‘semi coastal marine ecosystems’ areas is associated with the words “shore/beach”. According to CoastalWiki, a widely accessed public webpage curated by the Flanders Marine Institute, the beach or shore comprises the space from the mean low point of water tide to the line of permanent vegetation. This area includes the supralittoral zone fully or in part. In my opinion, the issue could be overcome by eliminating ‘shore/beach’ from the definition and only leaving the intertidal zone, as anything above its higher margin that is influenced by marine environment is considered supralittoral.

The proposed ‘coastal marine ecosystems’ zone present an issue in its definition. In this manuscript, it is initially indicated as the area covering the littoral zone, and in a second paragraph indicated as “exclusively submerged communities in seawater”. The encyclopedia Britannica, in the section dedicated to marine ecology, define the littoral zone as the “marine ecological realm that experiences the effects of tidal and longshore currents”, including the intertidal subzone. In this case, presenting overlapping with the ‘semi coastal marine ecosystems’ area. The definition of this area should be more specific in my opinion.

Ultimately, the difference between ‘coastal marine ecosystem’ and ‘deep marine ecosystem’ is not very clear: the definition of a specific depth or other characteristics to differentiate between the two zones are not indicated. The authors indicate “sea sediments, whale falls, and hydrothermal vents in the abyssopelagic zone” as habitats within the ‘deep marine ecosystem’, however, many other habitats are present in the abyssopelagic zone. Moreover, it is not very clear how should be classified the offshore deep-sea water column.

Saline lakes; are they marine ecosystems?

The paragraph reads well and has meaningful content. It is not clear what is the authors’ ultimate opinion on the classification of saline lakes and marine ecosystems.

Coastal upwelling, oceanic currents, and nutrient-rich areas

The paragraph reads well and has meaningful content. Authors indicate some geographical locations with oceanic currents and major upwelling as the greatest for fungal biodiversity (e.g., Southeast Asia). It would be helpful to provide data on the number of samples processed or studies performed in each reported world region to understand if the number of species described is influenced by a larger number of studies performed in a specific area.

Major upwelling systems with extensive studies of marine fungi

Articles reporting fungal biodiversity from these areas are not discussed in the paragraph, in my opinion they should be included and discussed in association with the presented upwelling systems.

Major upwelling systems understudied for marine fungi

It is not clear if the areas here described were ever studied for fungal biodiversity. If data are present, the studies should be here cited.

Poorly studied regions with great potential to improve fungal studies

Two comments are reported in the pdf file for sentences not easily understandable.

a. Ecological sampling and HTS as a tool of integrated sciences

In my opinion, this subtitle does not reflect the written content. It is not clear to me what the authors intends as “tool of integrated sciences” for ecological sampling and HTS.

In this paragraph, authors highlight that “High resolution of microbial diversity to the species level can be obtained by NGS targeting the internal transcribed spacer (ITS) genomic region”. Even though the reported literature is relevant, there are numerous studies highlighting the limitation of ITS region to discriminate between different fungal species (see, for example: https://doi.org/10.4137/EBO.S653 ; or https://doi.org/10.1128/AEM.00626-21 ; or https://doi.org/10.1038/s41579-018-0116-y ). In my opinion, it is important to discuss this limitation in the submitted manuscript.

c. Metabolomics and its importance in chemotaxonomy and discovering novel com- pounds

This paragraph is quite confusing and does not read well. Chemotaxonomy and compounds discovery are discussed intermittently and not properly defined. I suggest adding a clear definition for chemotaxonomy and metabolomics, along with their use in fungal taxonomy, biotechnology or biodiscovery. Moreover, the sentence reporting technologies used in metabolomics should be written better (abbreviations in brackets and ‘-’ added where needed).

Author Response

Dear Editor and Reviewers,

Thank you for your letter and comments on our manuscript entitled “Ecological and oceanographic perspectives in future marine fungal taxonomy (Manuscript ID: jof-1940225)”.

Your comments and those of two reviewers were highly insightful and enabled us to greatly improve the quality of our manuscript (jof-1940225). We have studied these comments carefully and have made corrections.

All revisions based on all valuable suggestions of the reviewers on the manuscript were revised, recorrected and marked up using “yellow highlighted” in the manuscript file.

Furthermore, all responses to comments from two reviewers are indicated with ">>>>" in red and yellow texts in this file. We hope that the revisions in the manuscript and our accompanying responses will be sufficient to make our manuscript suitable for publication in Journal of Fungi (JOF).

Best regards,

Nattawut Boonyuen, PhD.

======================================================

Reviewer II:

In this manuscript, the authors provide a general update on the status of marine mycology without going into details of the currently known marine fungal biodiversity. The article is presented as a ‘concept paper’ highlighting the current technical state of the art of marine mycology in terms of biodiversity and research techniques currently implemented. In this manuscript is presented a new definition for marine areas often subjected to sampling for microbial isolation, in the effort to normalize the location of microbes in future studies and support geographical understanding of marine fungal biodiversity. Authors also analyze oceanic currents worldwide, they highlight areas where high biodiversity was recorded and indicate other areas where few studies were performed, providing suggestions on where additional research should be carried out.

The manuscript is well written and easy to read, the literature is up to date and the information reported on fungal biodiversity in relation to Oceanic currents are interesting. However, I have a series of concerns about several topics reported in the manuscript, and therefore suggest thorough revision before submission.

Abstract

 The abstract is of appropriate length and well written. I made some comments for the sentence ‘diversity of marine fungi in the world’s oceans and provides an integrated and holistic view of their ecological roles’. See PDF file

>>>> We have revised as suggested in Page 1 as follows: “………..the current knowledge of the marine fungal diversity and provides an integrated and comprehensive view of their ecological roles in the world’s oceans”.

  1. Introduction

 In the introduction, authors provide a nice update on the history of marine fungal research, the methods involved in their identification, their definition, biodiversity and sites of isolation. The space dedicated to this paragraph and the figures are appropriate. I only made a couple of comments in the initial part of the text and in the figure one, where typographic errors are present. See PDF file.

>>>> We have recorrected and revised as suggested in Page 2 as follows: “…..Hawksworth and Lücking [2] listed the species associated with bryophytes, algae, endophytic fungi inside vascular plants, tropical foliicolous and fungicolous fungi, mammalian guts, insect guts, and exoskeletons, on and in rocks, and deep-sea and ocean sediments to demonstrate the diversity of fungal environments”.

>>>> We have recorrected and revised as suggested in Page 2 as follows:“………the total number of global fungal species has been estimated based on different approaches.”

>>>> We have revised as suggested in Page 2 as follows:“……… However, all marine fungi are reported from different marine ecosystems”.

  1. Coastal, semi-marine, and marine habitats – how can they be defined?

 In the second paragraph on this paper, the authors propose new definitions for marine habitats where fungi have been documented, in the attempt to better categorize future studies and direct research on areas previously not considered marine (i.e., Pandanus).

The definition and delimitation of specific marine areas for microbial isolation is important to advance understanding on worldwide fungal distribution and avoid misunderstanding in future studies. Therefore, agreement on consensus definitions is essential. However, the proposed definitions suggested by the authors might not fully support their purpose as they are described with terms not specific enough and are characterized by partial overlapping.

The suggested definitions are:

  1. ‘costal terrestrial ecosystems’ covering the supralittoral zone
  2. ‘semi coastal marine ecosystems’ covering shore/beach including intertidal zone (areas “directly influenced by tides”)
  3. ‘coastal marine ecosystems’ covering the littoral zone
  4. ‘deep marine ecosystems’ covering the deep sea

>>>> We have revised as suggested in Page 5 as follows:“………  in the supralittoral zone are not direct exposure to water, but can be influenced by splash and the soil often expose to the sunlight.”

Overlap between the proposed ‘costal terrestrial ecosystems’ and ‘semi coastal marine ecosystems’ areas is associated with the words “shore/beach”. According to CoastalWiki, a widely accessed public webpage curated by the Flanders Marine Institute, the beach or shore comprises the space from the mean low point of water tide to the line of permanent vegetation. This area includes the supralittoral zone fully or in part. In my opinion, the issue could be overcome by eliminating ‘shore/beach’ from the definition and only leaving the intertidal zone, as anything above its higher margin that is influenced by marine environment is considered supralittoral.

>>>> Thank you so much for this issue, the main concern of the reviewer is overlapping of ecosystems. It is true, in nature clear boundaries for ecosystems cannot be found. Since distinct factors to separate the ecosystems, and we have removed "shore" in describing 'semi coastal marine ecosystems' as shown in Page 6 as follows: “…….to those that exist between the intertidal zone and are influenced directly by tides. These habitats expose to the sun during low tides and get inundated during high tides. “…………environments include beach woods, ……”

Also, we added some texts to improve as following here.

>>>>We have revised as suggested and added some key sentence in Page 6 as follows:“………that exist between the intertidal zone and the shore and- are influenced directly by tides. These habitats expose to the sun during low tides and get inundated during high tides”.

>>>> We have revised as suggested and added some key sentence in Page 6 as follows:“……… In addition, because of the continuous availability of water, the sediments on the ocean bottom contain a higher amount of moisture than those in terrestrial ecosystems”.

The proposed ‘coastal marine ecosystems’ zone present an issue in its definition. In this manuscript, it is initially indicated as the area covering the littoral zone, and in a second paragraph indicated as “exclusively submerged communities in seawater”. The encyclopedia Britannica, in the section dedicated to marine ecology, define the littoral zone as the “marine ecological realm that experiences the effects of tidal and longshore currents”, including the intertidal subzone. In this case, presenting overlapping with the ‘semi coastal marine ecosystems’ area. The definition of this area should be more specific in my opinion.

>>>> We have revised as suggested and added some key sentence in Page 6 as follows: “……Coastal marine ecosystems receive sunlight through the water column and, hence, can be considered as productive and diverse areas in the seas. Submerged seagrasses,…….…..”

>>>> We have revised as suggested and added some key sentence in Page 6 as follows: “…….. are located beyond the littoral zone; therefore, sunlight is a limiting factor. Sea sediments, whale falls, and hydrothermal vents in the abyssopelagic zone (Figs 3 and 5) are the habitats for fungi in this ecosystem”.

Ultimately, the difference between ‘coastal marine ecosystem’ and ‘deep marine ecosystem’ is not very clear: the definition of a specific depth or other characteristics to differentiate between the two zones are not indicated. The authors indicate “sea sediments, whale falls, and hydrothermal vents in the abyssopelagic zone” as habitats within the ‘deep marine ecosystem’, however, many other habitats are present in the abyssopelagic zone. Moreover, it is not very clear how should be classified the offshore deep-sea water column.

>>>> We have revised as suggested and added some key sentence in Page 6 as follows: “……..Though, in nature, ecosystems cannot be separated by clear boundaries; hence,…….”

Saline lakes; are they marine ecosystems?

The paragraph reads well and has meaningful content. It is not clear what is the authors’ ultimate opinion on the classification of saline lakes and marine ecosystems.

>>>> We have revised and added additional information as shown in Page 7 (the last paragraph) as follows:“……..Marine fungi and marine-derived fungi exist widely in marine ecosystems, particularly in saline lakes. This habitat is of great importance to studying their species composition, diversity, and relationships with environmental factors to protect and maintain ecosystem balance [13]. Furthermore, salt lakes are essential, biologically and mineral resource-rich lakes that have significant research value and occur on all continents, including Antarctica, the Caspian Sea, the Dead Sea, and many of the highest lakes, such as those in Tibet and on the Altiplano of South America [13].

>>>> We have revised and added additional information as shown in Page 7 as follows: “……..However, herein, we regard that treating taxa from saline lakes surrounded by land as marine taxa is not appropriate because they represent a distinct ecosystem in ecological perspective”.

Coastal upwelling, oceanic currents, and nutrient-rich areas

The paragraph reads well and has meaningful content. Authors indicate some geographical locations with oceanic currents and major upwelling as the greatest for fungal biodiversity (e.g., Southeast Asia). It would be helpful to provide data on the number of samples processed or studies performed in each reported world region to understand if the number of species described is influenced by a larger number of studies performed in a specific area.

>>>> In our dataset, we considered the number of species recorded in a location, rather than the number of publications from the country/region. The summary of number of species recorded per country/region can provided as a supplementary table (see attached file)

Major upwelling systems with extensive studies of marine fungi

Articles reporting fungal biodiversity from these areas are not discussed in the paragraph, in my opinion they should be included and discussed in association with the presented upwelling systems.

>>>> We have revised and added additional information as shown in Page 11 (the last paragraph) as follows: “…….. Based on the findings [219], the role of fungi in the processing of marine organic matter in the upwelling environment in the coast of Chile shows that the seasonal variation in fungal biomass in the water column reflects their capability to hydrolyze organic polymers. Therefore, the biomass and activity of fungi in this environment respond to the seasonal cycle of upwelling [219]. In the Benguela Upwelling System (Namibia), the results indicate that the phylogenetic diversity of fungi across redox-stratified marine habitats corresponds to functionally relevant mechanisms that support in structuring carbon flow from primary producers in marine microbiomes from the surface ocean to the subseafloor [220].

Major upwelling systems understudied for marine fungi

It is not clear if the areas here described were ever studied for fungal biodiversity. If data are present, the studies should be here cited.

>>>> We have revised as suggested and add more citations in Page 11 as follows: “There are few studies on the upwelling systems associated with marine fungi, such as citations here [219–222]”.

Poorly studied regions with great potential to improve fungal studies

Two comments are reported in the pdf file for sentences not easily understandable.

>>>> We have revised as suggested and make more easily to understand in Page 12 as follows:“……...Despite the availability of funding, experts, mycologists, researchers, and other key stakeholders are required to carry out a research project and continue marine fungi-related activities through global collaborations”.

>>>> We have revised as suggested and make more easily to understand in Page 12 as follows: “……...but marine fungi are less studied when compared to these ecological fungal groups, e.g., in Australia.”.

  1. Ecological sampling and HTS as a tool of integrated sciences

In my opinion, this subtitle does not reflect the written content. It is not clear to me what the authors intends as “tool of integrated sciences” for ecological sampling and HTS.

In this paragraph, authors highlight that “High resolution of microbial diversity to the species level can be obtained by NGS targeting the internal transcribed spacer (ITS) genomic region”. Even though the reported literature is relevant, there are numerous studies highlighting the limitation of ITS region to discriminate between different fungal species (see, for example: https://doi.org/10.4137/EBO.S653; or https://doi.org/10.1128/AEM.00626-21; or https://doi.org/10.1038/s41579-018-0116-y). In my opinion, it is important to discuss this limitation in the submitted manuscript.

>>>> We have revised as suggested and added more discussion in Page 13 as follows: “…….. Although the ITS region has demonstrated strong resolution power, resulting in species discrimination in the most of fungal taxa, a key disadvantage is its inadequate resolution in closely related species, species complexes involving cryptic species, or metabarcoding studies. As a result, secondary DNA barcodes are increasingly being applied for groups where ITS does not give sufficient precision [144,146].

  1. Metabolomics and its importance in chemotaxonomy and discovering novel com- pounds

This paragraph is quite confusing and does not read well. Chemotaxonomy and compounds discovery are discussed intermittently and not properly defined. I suggest adding a clear definition for chemotaxonomy and metabolomics, along with their use in fungal taxonomy, biotechnology or biodiscovery. Moreover, the sentence reporting technologies used in metabolomics should be written better (abbreviations in brackets and ‘-’ added where needed).

>>>> In the text file, we already given the definition of chemotaxonomy in Pages 14-15 as follows: “…This is a branch of mycology that studies chemical variation in fungal cells and uses chemical characteristics to classify fungal species based on distinct differences and similarities in their biochemical compositions; it is particularly useful in the systematic approach of mycological polyphasic taxonomy”.

>>>> We have revised as suggested in Page 15 “…….and quantitative analysis of small molecules (molecular weight < 1500 Da;……”

>>>>Also, we have revised the abbreviations in brackets and changed them as the full name in Page 15 as follows: “………. such as Mass spectrometry (MS) e.g., gas-chromatography mass spectrometry, liquid chromatography-mass spectrometry, Tandem mass spectrometry, Capillary electrophoresis mass spectrometry…”

>>>> We have revised in Pages 15-16. Additional discussion was added, and a section (concluding remarks) was switched to the bottom.

>>>> Based on the whole paper (Pages 1-27), a few typo errors have already recorrected and re-double checked, and related citations were included as yellow

Round 2

Reviewer 2 Report

The authors reviewed their manuscript according to several suggestions and provided comments to all my questions. I hereby submit a number of comments I still have on the final draft, while some comments are reported on the pdf file.

2. Coastal, semi-marine, and marine habitats – how can they be defined?

The authors responded to my questions and added some information about the distinction between habitats. I still have a serious doubt on the usefulness of these definitions that contrast with habitats definitions currently in use. If the aim of the authors is to suggest consensus terms that will be used in the future to classify the origin of samples, I think that such goal might not be accomplished as the definitions are covering areas too wide and still overlapping.

Saline lakes; are they marine ecosystems?

The authors added a new paragraph. The added text is not helpful in addressing my question reported in the previous review, and contains sentences with confusing statements. I suggest to improve it or eliminate it. Some comments in the pdf file.

The authors also added a sentence at the end of the paragraph. In my opinion, since this section subtitle is a question, the actual text should begin with the response to the question, followed by an explanation and description of their statement.

Coastal upwelling, oceanic currents, and nutrient-rich areas

The authors added a table with the number of species recorder at locations included in the studies they reviewed. This is not what I intended in my previous question. I highlighted that having an idea of the number of samples processed, or studies performed, would help in understanding if a higher number of species isolated from a determined location/geographical area is merely reported because of a higher number of studies were performed, or because there is an actual higher biodiversity.

Poorly studied regions with great potential to improve fungal studies

>>>> We have revised as suggested and make more easily to understand in Page 12 as follows:“……...Despite the availability of funding, experts, mycologists, researchers, and other key stakeholders are required to carry out a research project and continue marine fungi-related activities through global collaborations”.

>>>> We have revised as suggested and make more easily to understand in Page 12 as follows: “……...but marine fungi are less studied when compared to these ecological fungal groups, e.g., in Australia.”.

I think this statement requires some data to support it. You could compare the number of published studies performed in this Country on terrestrial, freshwater and marine fungi and highlight the findings.

Ecological sampling and HTS as a tool of integrated sciences

>>>> We have revised as suggested and added more discussion in Page 13 as follows: “…….. Although the ITS region has demonstrated strong resolution power, resulting in species discrimination in the most of fungal taxa, a key disadvantage is its inadequate resolution in closely related species, species complexes involving cryptic species, or metabarcoding studies. As a result, secondary DNA barcodes are increasingly being applied for groups where ITS does not give sufficient precision [144,146].

In my opinion, this is still not reflecting the reality as ITS does not discriminate between most fungal taxa. It does it for a minimal number of genera, and rarely for species in the genera most commonly isolated from the marine environment such as Aspergillus and Penicillium.

1. Metabolomics and its importance in chemotaxonomy and discovering novel com- pounds

The authors have improved this paragraph but still some confusion remains. The text begins with natural products drug discovery, to then move onto chemotaxonomy definition. This is followed by the description of organism chemistry methods for metabolomics, and then a section on chemotaxonomy is discussed again. It does not read well.

Author Response

Thank you for your sending us the additional comments and invaluable constructive suggestions concerning our manuscript as the minor revisions. Those comments are all valuable and very helpful for revising and improving our review. We have read comments carefully and have made corrections that we hope meet with approval.

The modified version of MS has been uploaded to the manuscript submission system. On behalf of my co-authors, we would like to express our great appreciation again to editor and reviewers. We hope that the revisions in the manuscript and our accompanying responses will be sufficient to make our manuscript suitable for publication in JOF.

Best Regards,

Nattawut Boonyuen (Corresponding author)

++++++++++++++++++++++++++++++++++++++++++++++++++++++++++

Dear Dr. Boonyuen,

Your manuscript has been reviewed by experts in the field and we request that you make minor revisions before it is processed further. Please revise your manuscript according to the reviewers' comments and upload the revised file within 5 days. Please click on the "Peer Review Reports" below to find the reviewers' comments and the version of your manuscript to be used for your revisions.

Reviewer II: Comments and Suggestions for Authors

The authors reviewed their manuscript according to several suggestions and provided comments to all my questions. I hereby submit a number of comments I still have on the final draft, while some comments are reported on the pdf file.

>>>Thank so much to the reviewer again for the great suggestion and for the very patient modification. The following is a point-to-point response to the comments as below in green texts based on this file and revised word file.

  1. Coastal, semi-marine, and marine habitats – how can they be defined?

The authors responded to my questions and added some information about the distinction between habitats. I still have a serious doubt on the usefulness of these definitions that contrast with habitats definitions currently in use. If the aim of the authors is to suggest consensus terms that will be used in the future to classify the origin of samples, I think that such goal might not be accomplished as the definitions are covering areas too wide and still overlapping.

 >>> Thank you to the reviewer II for these constructive comments. We somewhat agree with the comment of the reviewer. Traditionally, mostly fungi in every marine habitat were defined as “marine fungi” without giving an idea of the particular habitat. In reality, those marine habitats show distinguish differences in physico-chemical and biological parameters and their processes. Hence, we believe that keeping the word “marine” is too broad and divide the marine environment as several phases as described in the paper.  Ecosystem boundaries cannot be defined from a single line. However, habitats of fungi belong to the suggested ecosystems defined by us.

Saline lakes; are they marine ecosystems?

The authors added a new paragraph. The added text is not helpful in addressing my question reported in the previous review, and contains sentences with confusing statements. I suggest to improve it or eliminate it. Some comments in the pdf file

>>>Thank you to the reviewer II for these valuable comments. In page 7, a new paragraph was deleted, according to you suggestion.

The authors also added a sentence at the end of the paragraph. In my opinion, since this section subtitle is a question, the actual text should begin with the response to the question, followed by an explanation and description of their statement.

Thank you for the suggestion. We have changed the subtitle as ‘Taxa reported from saline lakes’. We think this subtitle would be appropriate to provide our opinion and conclusion.

Coastal upwelling, oceanic currents, and nutrient-rich areas

The authors added a table with the number of species recorder at locations included in the studies they reviewed. This is not what I intended in my previous question. I highlighted that having an idea of the number of samples processed, or studies performed, would help in understanding if a higher number of species isolated from a determined location/geographical area is merely reported because of a higher number of studies were performed, or because there is an actual higher biodiversity.

>>> Yes, we agree with your remarks. We did not analyze the number of studies per region vs the number of species recorded.  From the map (Figure. 6), we would like to show the distribution of the known marine spp. but not to compare the numbers from different regions. Furthermore, to understand whether these regions are located at coastal-upwelling, oceanic currents, and nutrient-rich areas. In the next subtitle, we briefly mention the reasons for restricting the research in certain areas (e.g., experts, funding etc.). So, obviously, these regions should report with higher number of species. Surprisingly, several of these regions (with facilities, funding, experts) are mainly located in coastal upwelling, oceanic currents, and nutrient-rich areas (e.g., Japan, Taiwan, and Malaysia). We do not know the exact reason why marine mycologists targeted these regions. :) Most probably, funding!

However, we agree that your point. The map (Fig. 6) is modified as below and shows the species records and removes the magnitude of the number of species in the revised file.

Poorly studied regions with great potential to improve fungal studies

>>>>We have revised as suggested and make more easily to understand in Page 12 as follows:“……...Despite the availability of funding, experts, mycologists, researchers, and other key stakeholders are required to carry out a research project and continue marine fungi-related activities through global collaborations”.

>>>> We have revised as suggested and make more easily to understand in Page 12 as follows: “……...but marine fungi are less studied when compared to these ecological fungal groups, e.g., in Australia.”.

I think this statement requires some data to support it. You could compare the number of published studies performed in this Country on terrestrial, freshwater and marine fungi and highlight the findings.

 >>> We greatly thank the suggestions. We have revised it in the sentence and add some data to support them in page 12.

Ecological sampling and HTS as a tool of integrated sciences

>>>> We have revised as suggested and added more discussion in Page 13 as follows: “…….. Although the ITS region has demonstrated strong resolution power, resulting in species discrimination in the most of fungal taxa, a key disadvantage is its inadequate resolution in closely related species, species complexes involving cryptic species, or metabarcoding studies. As a result, secondary DNA barcodes are increasingly being applied for groups where ITS does not give sufficient precision [144,146].

In my opinion, this is still not reflecting the reality as ITS does not discriminate between most fungal taxa. It does it for a minimal number of genera, and rarely for species in the genera most commonly isolated from the marine environment such as Aspergillus and Penicillium.

>>> We are sorry for clarification of ITS that some time not reflect to all fungal taxa. We have revised and added the sentence as well as some updated citations to support the paragraph as shown in page 13.

  1. Metabolomics and its importance in chemotaxonomy and discovering novel compounds

The authors have improved this paragraph but still some confusion remains. The text begins with natural products drug discovery, to then move onto chemotaxonomy definition. This is followed by the description of organism chemistry methods for metabolomics, and then a section on chemotaxonomy is discussed again. It does not read well.

>>> We are sorry to make it confusing. Thanks for reviewer’s patience and valuable suggestions, your suggestions made the paragraph more coherent and we have rearrangement. It was revised as in page 15.
